# Photothermally induced natural vibration for versatile and high-speed actuation of crystals

Yuki Hagiwara[1], Shodai Hasebe[1], Hiroki Fujisawa[2], Junko Morikawa[2], Toru Asahi[1,3] & Hideko Koshima [3] ✉

The flourishing field of soft robotics requires versatile actuation methodology. Natural vibration is a physical phenomenon that can occur in any material. Here, we report high-speed bending of anisole crystals by natural vibration induced by the photothermal effect. Rod-shaped crystal cantilevers undergo small, fast repetitive bending (~0.2°) due to natural vibration accompanied by large photothermal bending (~1°) under ultraviolet light irradiation. The natural vibration is greatly amplified by resonance upon pulsed light irradiation at the natural frequency to realise high frequency (~700 Hz), large bending (~4°), and high energy conversion efficiency from light to mechanical energy. The natural vibration is induced by the thermal load generated by the temperature gradient in the crystal due to the photothermal effect. The bending behaviour is successfully simulated using finite element analysis. Any light-absorbing crystal can be actuated by photothermally induced natural vibration. This finding of versatile crystal actuation can lead to the development of soft robots with high-speed and high-efficient actuation capabilities.

Natural vibration is a physical phenomenon during which a material continues to vibrate at a certain frequency (the natural frequency) upon the application of an external force. Additionally, when a periodic force is applied near the natural frequency, the vibrational motion is amplified by resonance. For example, during an earthquake, if the natural frequency of a shaking building matches the frequency of the earthquake, the building will resonate and shake violently, which may lead to collapse of the structure[1]. Equipment and devices must be designed so that resonance does not occur, otherwise they may be damaged or become less durable[2]. Most musical instruments, such as the guitar and saxophone, create sound (natural vibration) via the movement of strings or reeds and amplify the sound via resonance in a box or tube[3]. Thus, resonated natural vibrations are widely related to our daily life. However, the natural vibration has not been paid attention in terms of material actuation.

We have focused on crystal actuation because mechanically responsive molecular crystals have potential application in actuators and soft robots[4]. Over the past decade, many mechanically responsive crystals have been developed that display bending[5–7], twisting[8], jumping[9] and flagella-like motion[10]. Most of these motions have been induced via photoisomerisation. Phase transitions can also induce mechanical motions, such as expansion/contraction[11], bending[12], jumping[13] and locomotion[14]. Mechanical motion of molecular crystals have been reviewed in several publications[15–19]. However, photo-isomerisation and phase transitions are not versatile due to the limitation that they can only occur in certain crystals. Furthermore, photoisomerisation-induced motion tends to be slow and thick crystals do not actuate[19].

Recently, we reported high-speed photothermally driven crystal bending[20–23]. The photothermal effect is a phenomenon by which

[1]Graduate School of Advanced Science and Engineering, Waseda University, 3-4-1 Okubo, Shinjuku-ku, Tokyo 169–8555, Japan. [2]School of Materials and Chemical Technology, Tokyo Institute of Technology, 2-12-1 Ookayama, Meguro-ku, Tokyo 152-8550, Japan. [3]Research Organization for Nano & Life Innovation, Waseda University, 513 Wasedatsurumaki-cho, Shinjuku-ku, Tokyo 162-0041, Japan. ✉e-mail: h.koshima@kurenai.waseda.jp

thermal energy (heat) is produced by the nonradiative deactivation of photoexcited materials; it occurs in any light-absorbing crystal. However, photothermally driven bending is small, with a bending angle typically less than 0.5°. One strategy to enhance photothermal bending is to focus on crystals with a large thermal expansion coefficient. Hence β-phase crystal (**1β**)[24] of 2,4-dinitroanisole (**1**) (Fig. 1a), which has relatively large thermal expansion coefficient, was used in this study.

While observing the bending of **1β** crystals due to the photothermal effect upon ultraviolet (UV) light irradiation, we have unexpectedly discovered for the first time that small, fast natural vibration occurs while accompanied by a large photothermal bending. More surprisingly, when irradiated with a pulsed UV laser of the same frequency as the natural frequency, the bending angle due to the natural vibration is considerably amplified by resonance, achieving high-speed, large bending, and high energy conversion efficiency from light to mechanical energy. Simulation of this high-speed bending is also successful. Resonated natural vibrations can expand the potential and versatility of crystals as actuation materials.

## Results
### Physical properties of the crystals
Colourless hexagonal rod-shaped **1** crystals were obtained from saturated methanol solution by cooling in a refrigerator at -5 °C for -10 minutes (Fig. 1b). Differential scanning calorimetry (DSC) measurements confirmed that the obtained crystals were β-phase, exhibiting a reversible phase transition α ↔ β at −5.1 °C and −8.0 °C on heating and cooling, respectively (Supplementary Fig. 3), in close agreement with literature values[24]. X-ray crystallographic analysis at 20 °C further confirmed that the rod-shaped crystal belonged to the monoclinic crystal system and the space group $P2_1/n$ (Supplementary Table 1, Supplementary Fig. 1, and Supplementary Data 1), consistent with the reported β-phase structure[24]. The **1β** molecules were aligned in a herringbone motif along the $b$-axis due to the two-fold helical axis and π−π stacking of 3.588 Å along the $a$-axis on the (001) plane (Fig. 1d). This resulted in the formation of the (010) plane as the top face and the longitudinal direction along the $a$-axis (Fig. 1c). The temperature dependence of the lattice constants indicated that the $a$-axis length increased linearly from 3.972 Å at 10 °C to 4.021 Å at 60 °C; the thermal expansion coefficient of the longitudinal direction along the $a$-axis was calculated as 247 MK$^{-1}$ (Fig. 1e, Supplementary Fig. 2 and Supplementary Table 2), which is several times larger than that (average value: 71.4 MK$^{-1}$) of many molecular crystals[25].

The ultraviolet-visible (UV-vis) diffuse reflectance spectrum of powdered **1β** crystals measured at room temperature showed strong absorption in the UV region, with an absorption peak at 370 nm and a shoulder at 450 nm (Fig. 1f).

### Natural vibration induced by the photothermal effect
When a hexagonal rod-shaped **1β** crystal (length 6075 µm, width 151 µm, width of the top face 40 µm, thickness 105 µm; crystal III in Supplementary Table 5) fixed at one end was uniformly irradiated from the top with the UV laser (375 nm, 1456 mW cm$^{-2}$) (Fig. 2a), the crystal was significantly bent due to the photothermal effect. Surprisingly, smaller and faster repetitive bending accompanied this large photothermal bending both under UV irradiation and after irradiation ceased (Fig. 2d and Supplementary Movie 1). Note that the crystal tip did not move out of focus during deformation, confirming that the crystal bent in the same plane and did not bend out of plane (Fig. 2a). In this study, crystal bending was evaluated using the bending angle, defined as the difference in tip displacement between the two most offset positions (Fig. 2c).

Upon UV irradiation, the photothermally driven bending away from the light source quickly reached a bending angle of 0.85° in 9.4 ms, then gradually increased to 1.22° in 100 ms (Fig. 2a, d). This bending angle is greater than the 0.2°–0.5° previously reported for other crystals[20–23] due to the larger thermal expansion coefficient (247 MK$^{-1}$) of the **1β** crystal compared with those (80–130 MK$^{-1}$) reported for other crystals. After stopping the UV exposure at 100 ms, the crystal bent up quickly to 0.32° in 17 ms, then slowly returned to 0.23° in 100 ms (Fig. 2a, d). The photothermally driven bending data were fitted with an exponential curve (Fig. 2e), giving time constants ($\tau_{on}$, $\tau_{off}$) of 4.7 and 6.4 ms for UV light on and off conditions, respectively. The surface temperature, which was simultaneously monitored by an infrared (IR) camera, increased from 25.6 °C to 35.3 °C upon 100 ms of UV irradiation (Fig. 2b) and decreased to 33.1 °C in 100 ms after the UV irradiation was stopped (red curve in Fig. 2d). The maximum bending angle and the maximum surface temperature increased linearly with increasing UV intensity (Fig. 2h, Supplementary Fig. 8 and Supplementary Table 4).

The smaller, faster repetitive bending was extracted by fitting the photothermal bending curve (Fig. 2e) to provide the vibration-like bending profile (Fig. 2f). The small bending angle was -0.20° initially, then gradually decreased to a steady angle of 0.02° at 80 ms. When the UV irradiation was stopped, the bending angle immediately recovered to 0.18°, then attenuated to a steady 0.04° in 20 ms. Fourier transform analyses of the time profile of bending with and without UV irradiation revealed that the frequencies of the small vibration-like bending were both 390 Hz (Fig. 2g). The bending angle of the small vibration increased linearly with increasing UV intensity. In contrast, the frequency did not change and remained at 390 Hz for any UV laser intensity. This confirmed that the small vibration-like bending was the natural vibration (Fig. 2i, Supplementary Fig. 8 and Supplementary Table 4).

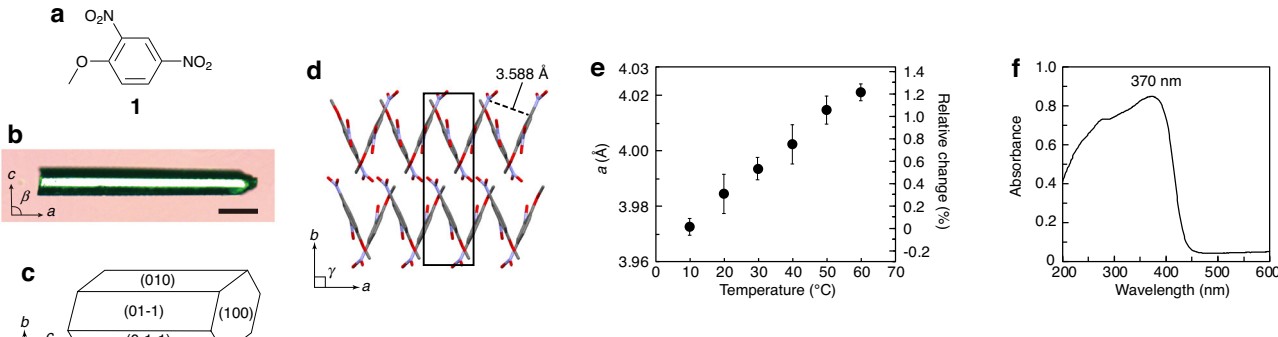

**Fig. 1 | Physical properties of the 1β single crystal. a** Chemical structure of 2,4-dinitroanisole (**1**). **b** A photo of a **1β** single crystal under crossed Nicols. The scale bar is 100 µm. **c** Face indices of the rod-shaped crystal. **d** Molecular arrangement on the (001) plane. Disordered oxygen atoms of two nitro substituents are omitted for clarity. **e** Temperature dependence of the $a$-axis length over the temperature range of 10–60 °C. Black bars indicate standard deviations. **f** UV-vis diffuse reflectance spectrum of powdered **1β** crystals.

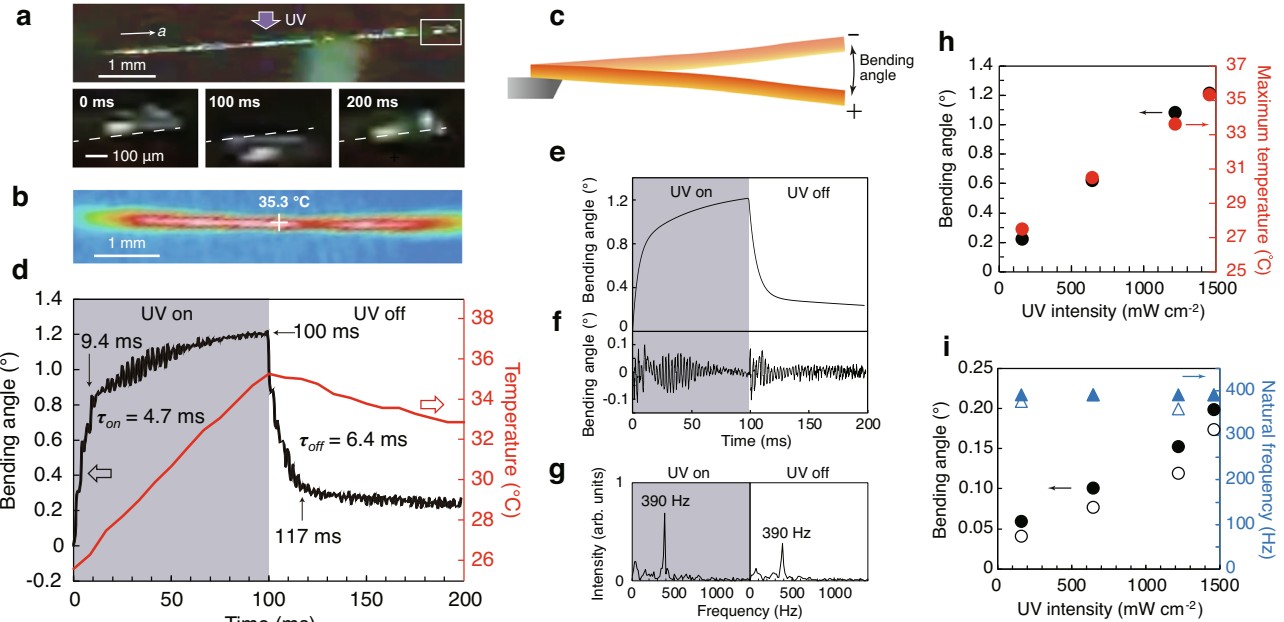

**Fig. 2 | Bending behaviour of 1β crystal III by the photothermal effect and the natural vibration (390 Hz) by irradiation with UV light (375 nm, 1456 mW cm⁻²).**
**a** Side view photo of a hexagonal rod-shaped crystal with the left end fixed to the glass needle, and the sequential snapshots of tip displacement before (left), upon (middle), and after (right) UV irradiation. **b** Temperature distribution of the irradiated crystal surface under UV irradiation for 100 ms. The cross mark indicates the measured temperature point. **c** Definition of the bending angle of the photothermal bending and the natural vibration. **d** Time dependence of the bending angle (black) and the maximum temperature (red) of the irradiated surface with and without UV irradiation for 100 ms. **e** Time profile of the fitted exponential curve of the large photothermally driven bending. **f** Time profile of the extracted bending angle of the small natural vibration. **g** Fourier transform analysis of (**f**) with and without UV irradiation. **h** UV intensity dependence of the photothermally driven bending angle (black circle) and the surface temperature (red circle) upon UV irradiation for 100 ms. **i** UV intensity dependence of the bending angle (black circle) of the natural vibration and natural frequency (blue triangle). Solid and open plots indicate the results upon UV irradiation and after UV cessation, respectively.

For a hexagonal rod-shaped **1β** crystal, the natural frequency $f_{cal}$ can be calculated according to Eq. (1):[1]

$$f_{cal} = \frac{h}{4\pi}\left(\frac{1.875}{l}\right)^2 \sqrt{\frac{E(b+3c)}{6\rho(b+c)}} \tag{1}$$

where $h$ is the thickness, $l$ is the length, $b$ is the width, $c$ is the width of the top surface, $E$ is the Young's modulus (1.65 GPa, Supplementary Fig. 7 and Supplementary Table 3) along the length direction and $\rho$ is the density (1.567 g cm⁻³, Supplementary Table 1). The calculated natural frequency $f_{cal}$ of 397 Hz was in close agreement with the measured value (390 Hz).

## Resonance amplification of a natural vibration

As mentioned above, the bending angle of the natural vibration was one order of magnitude smaller than that of the photothermally driven bending. Surprisingly, however, when crystal III was irradiated with pulsed UV light at the natural frequency of 390 Hz, the natural vibration was dramatically amplified by resonance (Fig. 3a, b and Supplementary Movie 2). The bending angle began at 0.20° in the first cycle and increased with irradiation time to the maximum of 3.4° after 150 ms; the difference in angle between the maximum of 2.4° and the minimum of −1.0° represented a 17-fold resonance amplification. Then, the resonated bending angle gradually decreased as the surface temperature increased. When the surface temperature reached a constant 40.8 °C after 850 ms under pulsed UV light, the resonated bending angle reached a steady value of 2.3° and maintained the angle with a quite small standard error of 0.003° during 850–1190 ms (133 cycles). Once the UV irradiation was stopped at 1190 ms, the bending angle rapidly decreased and the natural vibration almost disappeared within 200 ms. This resonated bending was observed for at least 460 cycles without any fatigue of the crystal under pulsed UV irradiation.

To determine the exact natural frequency of crystal III, we observed the bending behaviour while irradiating it with UV pulses of a wide range of frequencies; the natural frequency (the most amplified frequency) was 390 Hz (black open circle, Fig. 3c). In addition, the fitted natural frequency $f_{fit}$ determined by the forced vibration model (see the footnote in Supplementary Table 5) was also 390 Hz (red open circle, Fig. 3c).

Furthermore, bending amplification due to resonance was examined in detail using another crystal V with a higher natural frequency of 702 Hz (Supplementary Fig. 9 and Supplementary Table 5). The resonated bending angle increased linearly with intensity of the 702 Hz pulsed UV light (Fig. 3d and Supplementary Fig. 10). Interestingly, when UV light pulsed over a wide range of 5–1000 Hz was used, bending amplification by resonance was observed not only at the 702 Hz natural frequency $f$ but also at its odd fractions of 233 ($f$/3), 139 ($f$/5), 99 ($f$/7), 77 ($f$/9), 63 ($f$/11) and 53 ($f$/13) Hz (Fig. 3e), and various bending patterns were created (Fig. 3g–i and Supplementary Fig. 11).

This amplification was caused by the start of down-bending and up-bending due to the natural vibration at the same time when the UV light was turned on and off, respectively. To evaluate separately the photothermally driven bending and the natural vibration induced by pulsed UV irradiation of odd fractions of the natural frequency, the bending angle of the natural vibration was extracted by fitting the photothermally driven bending angle by the pulse frequency (Supplementary Fig. 12). The fitted line described a linear increase in bending angle of the natural vibration with increasing pulse frequency (blue, Fig. 3f). On the other hand, the photothermally driven bending angle decreased nonlinearly with increasing pulse frequency (red, Fig. 3f).

## Actuation performance

To clarify the relationship between crystal shape and natural vibration as well as photothermally driven bending, the bending behaviour of

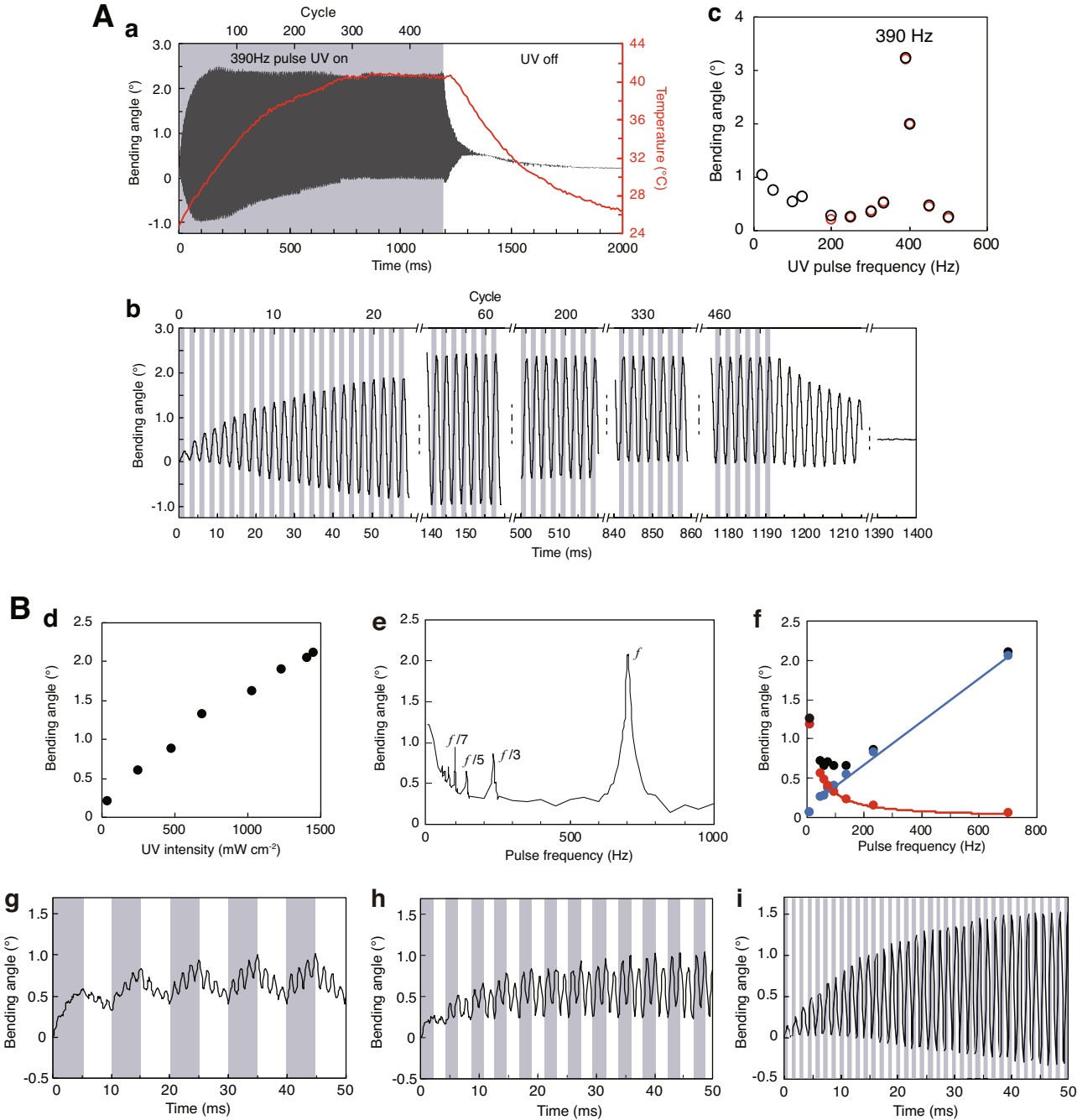

**Fig. 3 | Amplification of natural vibration by resonance. A 1β** crystal III with a natural frequency of 390 Hz. **a** Time profiles of the resonated natural vibration and surface temperature upon exposure to 390 Hz pulsed UV irradiation. **b** Partially enlarged view of **a**. **c** UV pulse frequency dependence of the measured (black) and fitted (red) amplified bending angles. **B 1β** crystal V with a natural frequency of 702 Hz. **d** 702 Hz pulsed UV intensity dependence of the maximum bending angle.

**e** UV pulse frequency (5–1000 Hz) relationship with the maximum bending angle. **f** Pulse frequency dependence of the measured bending angle (black), the photothermally driven bending angle (red) and the natural vibration (blue) at the odd fractions of the natural frequency (702 Hz). **g**–**i** Time profiles of high-speed bending upon pulsed UV irradiation of odd fractions of the natural frequency: **g** 99 (*f*/7), **h** 233 (*f*/3) and **i** 702 Hz (*f*).

five crystals (I–V) of different shapes were examined, with different geometries ranging from 5 to 8 mm in length and 50 to 220 μm in thickness (summarised in Supplementary Fig. 9 and Supplementary Table 5). When the crystals were continuously irradiated with UV light for 100 ms, they exhibited photothermally driven bending of 1.1°–1.7°, which increased slightly in proportion to the thickness (Fig. 4a). The natural vibrations of 0.06°–0.3° were extracted and were greatly amplified to 1.9°–4.0° due to resonance upon irradiation with pulsed UV light at each natural frequency; these were nearly proportional to the aspect ratio (length/thickness) of the crystals (Fig. 4b, c). The

natural frequencies were in the range of 200–700 Hz and were nearly proportional to thickness/length² ($h/l^2$), in accordance with Eq. (1) (Fig. 4d). Moreover, the resonance amplification ratio ranging from 9 to 32 also increased in proportion to the thickness/length² (Fig. 4e). Summarising, the frequency and the bending angle could be changed and tuned by changing the crystal shape; larger bending could be realised by resonating thinner and longer crystals, and conversely, the higher natural frequency could be realised by using thicker and shorter crystals (Fig. 4f). However, precise control over uniformity, dimensions, and shape of crystals is still challenging; it is an essential

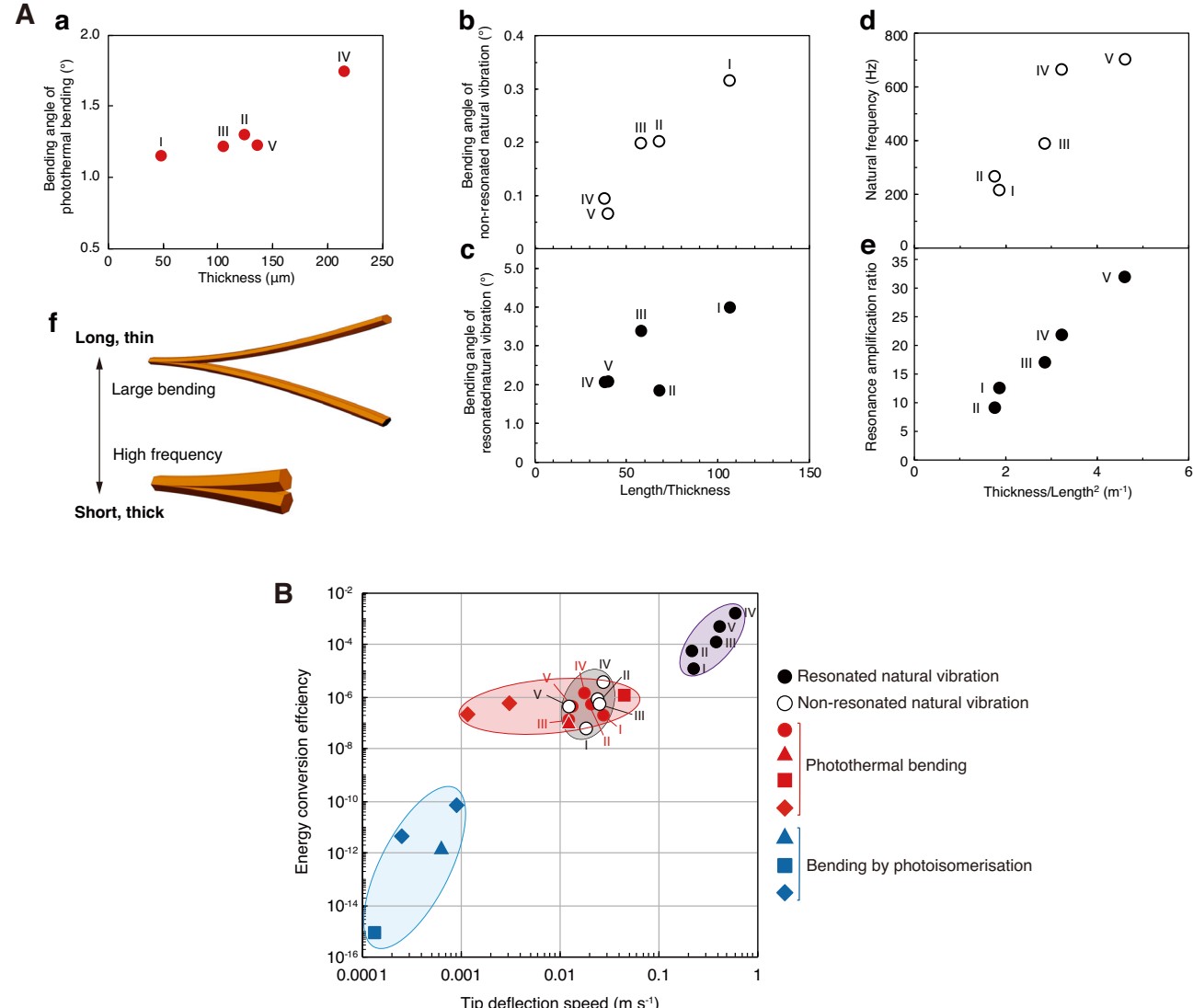

**Fig. 4 | Actuation performance. A** Relationship between crystal shape and bending behaviour of **1β** crystals I–V. **a** Thickness dependence of the photothermally driven bending angle. **b, c** Aspect ratio (length/thickness) dependence of the maximum bending angle of the non-resonated natural vibration (**b**) and the resonated natural vibration (**c**). **d, e** Thickness/length² dependence of the natural frequency (**d**) and the resonance amplification ratio (**e**). **f** Schematic diagram of crystal shapes to realise large bending angles and high frequencies. **B** Relationship between

the energy conversion efficiency and the tip deflection speed: the resonated natural vibration (black solid circle in purple ellipse), the non-resonated natural vibration (black open circle in grey ellipse), and the photothermally driven bending (red circle in red ellipse) of 1β crystals I–V; the photothermally driven bending (red square[21], red triangle[22], and red rhombus[23] in red ellipse) and the photoisomerisation driven bending (blue square[21], blue triangle[22], and blue rhombus[23] in blue ellipse) of other crystals.

requirement toward the development of crystal actuators with the desired output.

Next, we evaluated the tip deflection speed of **1β** crystals I–V and compared the results with other photomechanical crystals (Fig. 4b and Supplementary Table 6). The tip deflection speed $v_l$ of the resonated natural vibration was in the range of 0.2–0.6 m s⁻¹ (black solid circle), which was more than ten times faster than the 0.01–0.03 m s⁻¹ (black open circle) of the non-resonated natural vibration. The tip deflection speed (0.001 – 0.05 m s⁻¹) by the photothermal effect itself of the **1β** crystal (red circle) and other crystals (red triangle[21], square[22], and rhombus[23]) is slightly slower to comparable to that of the non-resonant vibration, but still fast because the photothermal effect takes place at the picosecond lavel[21, 22] in the primary photophysical process. On the other hand, the tip deflection speed by photoisomerisation (blue triangle[21], square[22], and rhombus[23]) is slow (0.0001 – 0.001 m s⁻¹) because photoisomerisation takes place at the nanosecond lavel[21, 22] in the secondary photochemical process. To the best of our knowledge,

the bending speed of the resonated natural vibration reported herein is the fastest of reported mechanically responsive crystals. The larger Young's modulus of the **1β** crystal than those of polymers and gels realised the high natural frequency (700 Hz), contributing the fast tip deflection speed.

The energy conversion efficiency $\eta$ from incident light energy $U_L$ to mechanical energy $U_M$ was evaluated according to Eq. (2):[26]

$$\eta = \frac{U_M}{U_L} = \frac{\frac{1}{2}m_e v_l^2}{I_L lbt_I} \tag{2}$$

where $m_e$ is the effective weight of the crystal ($m_e = \frac{104}{405}m$ where $m$ is the crystal weight), $I_L$ is the UV irradiation intensity per area, and $t_I$ is the irradiation time. The energy conversion efficiency of the resonated natural vibration was 10⁻⁵–10⁻³ (black solid circle), which is two-to-three orders higher than that (10⁻⁸–10⁻⁶) of the non-resonated natural vibration (black open circle) and the 10⁻⁷–10⁻⁶ of the photothermally driven bending (red circle). Compared with other crystals, this

efficiency is two and three orders of magnitude higher than the $10^{-7}$–$10^{-6}$ of bending by the photothermal effect (red triangle[21], square[22], and rhombus[23]) and ten orders of magnitude higher than that the $10^{-15}$–$10^{-10}$ of photoisomerisation (blue triangle[21], square[22], and rhombus[23]). Thus the resonated natural vibration induced actuations with the fastest speed and the highest energy conversion efficiency.

## Discussion

Photothermal bending is caused by the non-steady thermal elongation difference between the irradiated and back surfaces, which is due to the non-steady temperature gradient formed by the heat conduction of photothermal energy[21]. On the other hand, the natural vibration is caused by the thermal load in the crystal generated by the photothermal effect. Despite these different bending mechanisms, we attempted to simulate actual bending behaviour by combining both photothermally driven bending and natural vibration.

Figure 5a presents a schematic illustration of the geometrical parameters of a hexagonal crystal cantilever for photothermally driven bending and natural vibration. Upon photoirradiation, heat is generated at the irradiated surface by the photothermal effect, conducting into the thickness direction according to the non-steady heat conduction equation, Eq. (3):[21]

$$\frac{\partial T(y,t)}{\partial t} = \alpha \frac{\partial^2 T}{\partial y^2} \qquad (3)$$

where $\alpha$ is the thermal diffusivity ($1.34 \times 10^{-7}$ m$^2$ s$^{-1}$, Supplementary Fig. 4) along the thickness direction, $y$ is the displacement in the thickness direction, $t$ is time and $T(y, t)$ is the temperature at a certain $y$ and $t$. Consequently, the temperature gradient and the thermal load are generated and trigger the photothermal bending and the natural vibration, respectively.

In the elastic cantilever system, the photothermal bending and the natural vibration are explicitly included in Eq. (4):[27]

$$F(x,t) = \rho A \frac{\partial^2 \delta(x,t)}{\partial t^2} + EI \frac{\partial^4 \delta(x,t)}{\partial x^4} \qquad (4)$$

where $A$ is the cross-sectional area, $x$ is the displacement along the length direction, $\delta(x, t)$ is the deflection at a certain $x$ and $t$ and $F(x, t)$ is the applied thermal load at a certain $x$ and $t$ derived from the temperature gradient. Using the tip deflection $\delta(l, t)$ calculated according to Eq. (4), the combined bending angle $\theta$ of the photothermal bending and the natural vibration can be calculated using Eq. (5):

$$\theta(t) = \frac{\delta(l,t)}{l} \qquad (5)$$

We simulated the bending angle by finite element analysis (FEA) using the ANSYS software package[28]. The temperature gradient was calculated by non-steady heat conduction analysis, and then the actual bending was simulated by coupled thermo-structural analysis (Supplementary 6 and Supplementary Fig. 13).

We conducted two types of simulations as described below.

### Simulation I

The bending simulation of crystal III was conducted based on the surface temperature changes measured by infrared (IR) thermography. The measured temperature was applied to the top surface, while the temperatures of two upper slanting surfaces were set according to the fitted temperature distribution (Supplementary 6.1 and Supplementary Fig. 14). According to Eq. (3), the temperature gradient was formed in the thickness direction and the temperatures of the irradiated and the back surfaces after UV irradiation for 100 ms were calculated to be 35.8 °C and 32.6 °C, respectively (Fig. 5b, c), giving a temperature difference of 3.2 °C (Fig. 5d).

Coupled thermo-structural analysis was then performed based on the calculated temperature gradient. Due to the one-directional temperature gradient in the thickness direction (Fig. 5b), the bending deformation was also confirmed to be in the vertical direction. The bending angle and time constant of the photothermal bending were well simulated to be 1.26° and 4.2 ms, respectively, which are comparable to the 1.22° and 4.7 ms of the measured bending (Fig. 5e and Supplementary Movie 3). The 390 Hz non-resonated natural vibration was also successfully simulated.

Next, the resonated natural vibration under 390 Hz pulsed UV irradiation was simulated. First, the time dependence of the temperature difference between the irradiated and back surfaces was calculated (Fig. 5f). Coupled thermo-structural analysis was then performed; the calculated time profile of amplified bending angle was nearly coincident with the measured one (Fig. 5g and Supplementary Movie 4).

### Simulation II

The bending simulation of crystal III, based on the irradiated light energy, was also performed, without the surface temperature measurement. When the upper surface of the crystal was irradiated with UV light (375 nm, 1456 mW cm$^{-2}$), almost all of the irradiated light energy should have been absorbed by the crystal because this **1β** crystal emitted very weak fluorescence (quantum yield: 0.005) (Supplementary Fig. 6). The specific heat capacity (1.53 J g$^{-1}$ K$^{-1}$, Supplementary Fig. 5) was used to translate heat energy to temperature. Simulation II is detailed in Supplementary 6.2 and Supplementary Fig. 15. The simulated and measured time profiles for bending under UV irradiation for 100 ms were comparable (Fig. 5h and Supplementary Movie 5). Furthermore, the simulated profile of amplified, repeated bending under 390 Hz pulsed UV irradiation was completely coincident with the measured profile (Fig. 5i and Supplementary Movie 6). Thus, the bending simulation II based on the irradiated light energy was successful, even without using the measured temperature as in Simulation I.

We have successfully simulated photothermal-induced natural vibration based on FEA as shown in Simulation I and II. In the past decade, a few models have been proposed to reproduce photomechanical bending of crystals through photoisomerisation−chemical reactions of molecules−based on rigorous mathematical equations[29,30]. The aim of these mathematical models is to understand the photoreaction of molecules in the crystalline state from the perspective of solid-state photochemistry. On the other hand, one of the objectives of this study is to reproduce high-speed bending of crystals through the photothermal effect and natural vibration−fundamental physical phenomena−and elucidate the mechanism of high-speed bending based on FEA models. Such FEA models are expected to enable a deeper understanding of mechanical motions of crystals from the viewpoint of not only materials chemistry but also materials engineering.

In conclusion, we discovered that molecular crystals cause high-speed natural vibration by the photothermal effect. The natural vibration was dramatically amplified by resonance to achieve high-speed, large bending, and high energy conversion efficiency. Any light-absorbing crystal can be actuated by photothermally induced natural vibration. This versatile and fast crystal actuation is expected to contribute to the development of soft robotics with high-speed and high-efficient actuation capabilities.

## Methods

### Material preparation and characterisation

Commercially available **1** (TCI) was used as-received, without further purification. A sample (300 mg) was dissolved in MeOH (4 mL) with heating, and the resultant solution was cooled in a refrigerator at 5 °C for a few tens of minutes to obtain colourless rod-shaped crystals.

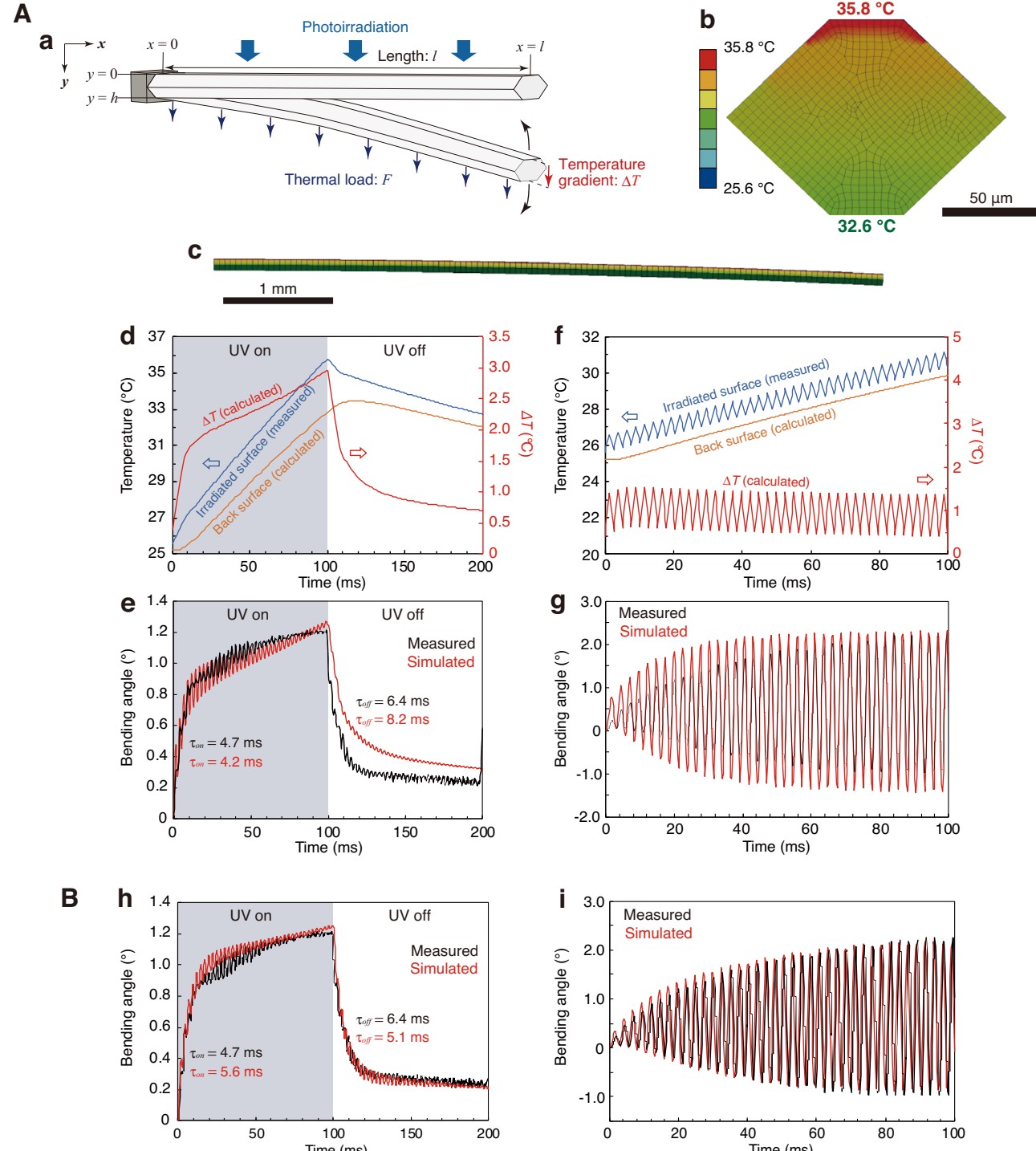

**Fig. 5 | Simulation of 1β crystal III bending by the combination of the photothermal effect and the natural vibration. A** Simulation I based on the measured crystal surface temperature. **a** Schematic illustration of crystal bending.
**b**, **c** Simulated temperature distribution after UV laser irradiation for 100 ms: the (100) cross-section (**b**) and the side view of the bent crystal (**c**). **d** Time dependence of the top surface (blue, measured) and back surface (orange, calculated) temperatures and the temperature difference between the two surfaces (ΔT, red, calculated) with and without UV light irradiation for 100 ms. **e** Time profiles of measured (black) and simulated (red) bending angles with and without UV light

irradiation for 100 ms. The inset values of $\tau_{on}$ and $\tau_{off}$ indicate the time constants for bending and straightening. **f** Time dependence of the top surface (blue, measured) and back surface (orange, calculated) temperatures and the temperature difference between the two surfaces (ΔT, red, calculated) under 390 Hz pulsed UV irradiation. **g** Time profiles of measured (black) and simulated (red) bending angles, amplified under 390 Hz pulsed UV irradiation. **B** Simulation II based on the irradiated light energy. **h** Time profiles of measured (black) and simulated (red) bending angles with and without UV light irradiation for 100 ms. **i** Time profiles of measured (black) and simulated (red) bending angles, amplified under 390 Hz pulsed UV irradiation.

DSC thermograms of **1β** crystals were recorded using a DSC8500 instrument (Perkin Elmer) with an empty aluminium pan as a reference. Enthalpy changes during thermal phase transitions were determined during heating and subsequent cooling between −50 °C and 120 °C. The same instrument was used to measure the specific heat capacity during heating.

Single-crystal X-ray diffraction data of crystals of **1β** at 20 °C were collected using an R-AXIS RAPID diffractometer (Rigaku) equipped with a monochromatic Mo-Kα radiation source ($\lambda$ = 0.71069 Å) operating at 50 kV and 40 mA. The temperature of the sample was regulated using a nitrogen gas flow cryostat. The crystal structure was solved using the integrated space group and crystal structure determination algorithm of SHELXT[31] software and refined on $F^2$ using the full-matrix least-squares SHELXL method[32]. The calculation was performed using the Olex2 molecular graphical interface[33]. Face indices and unit cell constants over the temperature range of 10−60 °C were measured using the same equipment.

UV-vis diffuse reflectance spectrum of powdered **1β** crystals was measured using a spectrophotometer (Lambda 650; PerkinElmer) equipped with an integrating sphere. The powder was mounted on a glass plate and then covered with a quartz plate. The experiment was performed over the wavelength range of 200−800 nm at 1 nm resolution, using a scan speed of 266.75 nm min$^{-1}$.

Fluorescence spectrum ($\lambda_{ex}$ = 300 nm) of powdered **1β** crystals was measured using a spectrofluorometer (FP-6500; JASCO). The fluorescence quantum yield ($\lambda_{ex}$ = 375 nm) of the **1β** single crystals was determined with an absolute photoluminescence quantum yield spectrometer (C9920-02G; Hamamatsu Photonics).

Thermal diffusivity of the **1β** single crystal was measured by the TWA technique[34] along the thickness direction (*b*-axis) at room temperature (25 °C). A single crystal was positioned between an indium−tin oxide glass heater and a nickel−gold microthermocouple sensor. The thermal diffusivity was calculated based on the relationship between the phase delay of the temperature waves between the heater and the sensor, and the frequency of the temperature wave.

Young's modulus of the **1β** crystal was measured using a universal testing machine (RTG-1210; A&D). The displacement dependence of applied load was measured by cantilever bending test.

## Observation of crystal bending

Photothermally driven bending of a rod-shaped **1β** crystal was observed using a digital high-speed microscope (VW-6000; Keyence). The entire rod-shaped crystal was irradiated using a UV laser ($\lambda$ = 375 nm; FOLS-03; SAWAKI Kobo) equipped with a fiber line generator (FLG10FC-633; Thorlabs). The surface temperature distribution was recorded using an IR thermography system (FSV-2000; Apiste). After collecting thermography images, the emissivity of a **1β** crystal was modified by applying a black-body coating (emissivity: 0.96; TA410KS; TASKO) to the rod-shaped crystal and the surface temperature was calibrated using the modified crystal emissivity of 0.95. The bending motion and surface temperature distribution were recorded at 8,000 and 120 fps, respectively. Various UV pulse frequencies were generated as square waves (duty ratio: 50%) via transistor-transistor logic control of the UV laser by an Arduino Leonard microcomputer device. The bending angle of the crystal was automatically analysed using Tracker, a video analysis and modelling tool[35].

## Bending simulation

Finite element analysis simulations of the photothermally driven bending were performed using ANSYS software[28]. A simplified three-dimensional (3D) analysis geometry of the single crystal was created using SpaceClaim direct 3D solid modelling software[28]. Material properties of **1β** crystals were assigned to the structure. The 3D geometry of the single crystal was divided into elements via the meshing

process. The detailed computational process is described in Supplementary 6.

## Data availability

Crystallographic data for the structures reported in this Article have been deposited at the Cambridge Crystallographic Data Centre, under deposition numbers CCDC 2192677 (**1β**). Copies of the data can be obtained free of charge via https://www.ccdc.cam.ac.uk/structures/. The other data that support the findings of this study are available from the corresponding author upon request.

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

## Acknowledgements

This research was supported by JSPS Grant-in-Aid for Scientific Research B (17H03107) for H.K., JSPS Research Fellowship for Young Scientists (21J20125) for Y.H., and the Grant-in-Aid for Young Scientists (Early Bird) at Waseda Research Institute for Science and Engineering for Y.H. We thank Dr. Meguya Ryu for the help of thermal diffusivity measurement. We also thank Mr. Ryutaro Yamaguchi for the help of UV pulse generation using Arduino Leonard. Y.H. and S.H. thank the Graduate Program for Power Energy Professionals in Waseda University.

## Author contributions

Y.H. and H.K. designed the experiments and wrote the paper. Y.H. performed all experiments except Young's modulus and fluorescence quantum yield, which were measured by S.H. H.F. and J.M. assisted with the thermal diffusivity measurements. T.A. assisted with the project. H.K. conceived the project.

## Competing interests

The authors declare no competing interests.
