## [Peer Review File · Nature Communications]

Photothermally induced natural vibration for versatile and high-speed actuation of crystalsReviewers' Comments:

Reviewer #1:

Remarks to the Author:

The authors studied the effect of external radiation and resonance with the natural frequency of a crystal of anisole derivative which were selected for having a large thermal expansion coefficient. This is a carefully performed study that goes beyond the chemistry and into mechanical engineering. Nevertheless, it brings a very important aspect to the future application of organic crystals as actuators and therefore it is of general interest and suitable for publication in Nature Communications. I have several comments for the authors.

Abstract

element analyses --- > element analysis

In the introduction, it says "high conversion efficiency". Do the authors refer to the elastic energy or the energy of the light used for resonance? This should be specified.

The last sentence of the abstract is too general and perhaps should be more specific, because soft robotics is already a developed research field.

In the abstract, please state typical bending angle range for this crystal (in degrees). It is important to note that these bending angles are very small compared to photoinduced bending.

Main text

Please mention early in the text how is the bending angle defined: is it the difference between the two most offset positions, or between the original position and the deflection to one side? The latter would be half of the former angle. This is shown in Figure 2c, but it should be mentioned in the text for clarity.

As shown in Figure 2a, the tip of the crystal is not very clear. How was the tip position determined, given the very fast vibration of the crystal? Please add standard deviations to the angle so that the precision of the measurement is clear.

p. 4. Bend angle - bending angle. Please also change this throughout the text.

Figure 1e: please add the standard deviations for each point or make a note in the caption if they are smaller than the symbols

p. 6. "much greater" is probably an overestimation, it's better to state just "greater". All of these changes are very small.

Please confirm whether the crystal during bending remained within the plane or not. This may affect the measured angle from the recorded videos.

p. 6. "infrared camera" and "thermography" have the same meaning. Please delete one of them.

p. 7. How was the Young's modulus along the long direction measured?

Perhaps in the discussion the authors can add two comments for the readers who are not familiar with mechanical engineering:

(1) What would one need to do to increase the bending angle?

(2) What would one need to do to increase the frequency of vibration?

The quality of the images in panels d and c in Figure 1 should be increased.

Please comment on how does the model for the photothermal bending relate to the recent quantitative model developed for photoinduced bending (Chem. Sci., 2018, 9; 2319–2335, Inorg. Chem. (2022) 61, 3573–3585) and include these related references.

Reviewer #2:

Remarks to the Author:

In this manuscript, Koshima et al. reported vibration of single crystals of 2,4-dinitroanisole induced by photoirradiation. The authors have been focusing on the crystalline compounds, which show large lattice parameter changes upon temperature change, and demonstrated the advantage of this strategy for development of photo-triggered actuating materials. In fact, the authors have already reported photothermally induced bending motion of single crystals, which also involved photoisomerization of component molecules and/or phase transitions during the mechanical motion.

Unlike the previous systems developed by the authors, the crystalline system presented in this manuscript is the simplest one, where the component molecules only absorb UV-lights to generate the excited state, and neither photoisomerization nor phase transition occurs during the bending motion of the crystal. In addition, by taking advantage of pulsed light to cause the resonance with the natural frequency, amplification of the bending angle and fast actuating motion could be achieved. The methodology presented here seems to be applicable to wide variety of compounds, which allow for the photothermal process, and would give strong impacts on the field of organic functional materials. On the other hand, there are mainly three crucial issues which were not mentioned in the contents of the presented study.

1) Organic crystals are three-dimensionally anisotropic due to the unsymmetric shape of the molecules. This means that physical parameters, such as Young's modulus, thermal diffusivity, as well as natural frequency, will have different values along different directions of the crystal. In the present study, the authors irradiated the (010) plane of a single crystal and bending of the crystal is described to occur parallel to the direction of the irradiated light, that is, perpendicular to the irradiated face. However, the shape of the crystal is indeed hexagonal, and (01-1) plane would also be irradiated. If the Young's modulus in the direction perpendicular to the (010) plane is not the smallest, and/or if the thermal diffusivity is larger along the direction perpendicular to the (01-1) plane than that perpendicular to the (010) plane, the actual motion of the crystal may be more complexed than a simple linear bending motion. In fact, how large are the differences in the above-mentioned physical parameters among those along the different directions in a single crystal, and is it collect that the observed bending motion is simple as described in Fig. 5Aa?

2) The authors claim that the bending speed and the resonated natural vibration in this system is the fastest among the previously reported systems. It is necessary to explain why such high performance could be achieved in the present system. In other words, does 2,4-dinitroanisole have any specific feature in the crystal packing which allows for such high performances? What is the most important parameter for achieving these?

3) What are the advantages of using the photothermal effects over IR irradiation? IR might be more energetically efficient than photoexcitation and would be applicable to much wide variety of compounds.

Reviewer #3:

Remarks to the Author:

Publication is recommended after a revision. The first paragraph of the introduction "Natural vibration ... daily life" is general knowledge and can be deleted. The manuscript should be edited. Please define "by fast cooling". Please explain why the X-ray crystallographic analysis were carried at room temperature and not under LN2 conditions as is common. What is the origin of the strong absorption in the UV region (370 nm and a shoulder at 450 nm)? For how long are these crystals stable under UV light? PXRD measurements should be included. There is no need to justify this work with possible applications such as soft robotics. Data is shown for bend angles with UV on and UV off; what happens when the crystals are again exposed to UV? The authors should provide an indication of the errors of their experiments (temperature, bend angle and so on). What is a polarised photo (Figure 1b)? Although videos are provided, I would like to have a few clear photos included in the main text supporting cartoons as shown in Fig 2c. A possible drawback of this work is the need for the formation of highly uniform crystals. As the authors showed, a different crystal shape results in different frequencies and bend angles. This effect limits the potential use of such crystals in possible applications. There is no need to justify this work with possible applications such as soft robotics. Instead, I would recommend that the authors address/discuss the issue of the need for development of crystal methods with control over crystal uniformity, dimensions and shape. I also recommend a discussion why these crystals have better properties than other systems

Response to the comments from the reviewers

We thank the reviewers for the good evaluation and valuable comments of our paper and recommendation to publish after revisions. We have thoroughly revised the manuscript according to the reviewers' comments.

Response to the comments from Reviewer 1

Comments: The authors studied the effect of external radiation and resonance with the natural frequency of a crystal of anisole derivative which were selected for having a large thermal expansion coefficient. This is a carefully performed study that goes beyond the chemistry and into mechanical engineering. Nevertheless, it brings a very important aspect to the future application of organic crystals as actuators and therefore it is of general interest and suitable for publication in Nature Communications. I have several comments for the authors.

Response: Thank you for your good evaluation and your informative comments. According to your comments, we have revised the manuscript and the figures.

Comment 1: Abstract : element analyses --- > element analysis

Response 1: "finite element analyses" in the Abstract is corrected to "finite element analysis".

Comment 2: In the introduction, it says "high conversion efficiency". Do the authors refer to the elastic energy or the energy of the light used for resonance? This should be specified.

Response 2: Energy conversion efficiency was estimated from irradiated light energy to mechanical energy, according to equation (2). In the abstract and introduction, "from light to mechanical motion" is added behind "high conversion efficiency".

Comment 3: The last sentence of the abstract is too general and perhaps should be more specific, because soft robotics is already a developed research field.

Response 3: We understand that the last sentence of abstract is too general. According to your suggestion, the last sentence of the abstract is corrected as follow:

[Before] "This finding of versatile crystal actuation can lead to the development of soft robotics."

[After] “This finding of versatile crystal actuation can lead to the development of soft robots with high-speed and high-efficient actuation capabilities.”

Also, the last sentence of “Discussion” section is corrected as below:

[Before] “This versatile and fast crystal actuation is expected to contribute to the development of soft robotics.”

[After] “This versatile and fast crystal actuation is expected to contribute to the development of soft robotics with high-speed and high-efficient actuation capabilities.”

Comment 4: In the abstract, please state typical bending angle range for this crystal (in degrees). It is important to note that these bending angles are very small compared to photoinduced bending.

Response 4: Bending angles are added in the abstract as follow, according to Supplementary Table 2:

“Rod-shaped crystal cantilevers undergo small, fast repetitive bending ($\sim 0.2^\circ$) due to natural vibration accompanied by large photothermal bending ($\sim 1^\circ$) under ultraviolet light irradiation. The natural vibration is greatly amplified by resonance upon pulsed light irradiation at the natural frequency to realise high frequency (~ 700 Hz), large bending ($\sim 4^\circ$), and high energy conversion efficiency from light to mechanical energy.”

Comment 5: Main text : Please mention early in the text how is the bending angle defined: is it the difference between the two most offset positions, or between the original position and the deflection to one side? The latter would be half of the former angle. This is shown in Figure 2c, but it should be mentioned in the text for clarity.

Response 5: The bending angle is defined as the difference between the two most offset positions. Thus, the following sentence is added to the end of the first paragraph in “Natural vibration induced by the photothermal effect” section.

“In this study, crystal bending was evaluated using the bending angle, defined as the difference in tip displacement between the two most offset positions (Fig. 2c).”

Comment 6: As shown in Figure 2a, the tip of the crystal is not very clear. How was the tip position determined, given the very fast vibration of the crystal?

Response 6: The movement of the crystals was captured by a high-speed camera (8000 frames per second) and the tip of the crystal on each frame were automatically tracked with a video analysis and modelling tool, Tracker. Therefore, the bending of the crystals was accurately measured. But as you commented, Fig 2a was unclear, so we have

changed it to a clearer image.

Comment 7: Please add standard deviations to the angle so that the precision of the measurement is clear.

Response 7: We observed crystal bending only once in each condition. We cannot add standard deviations.

Comment 8: p. 4. Bend angle - bending angle. Please also change this throughout the text.

Response 8: “bend angle” in the text is completely corrected to “bending angle” throughout the text, and the same modification is conducted to figures in the revised manuscript.

Comment 9: Figure 1e: please add the standard deviations for each point or make a note in the caption if they are smaller than the symbols

Response 9: The standard deviations are added in Fig. 1e, and the following sentence is added in the legend of Fig. 1.

“Black bars indicate standard deviations.”

Comment 10: p. 6. “much greater” is probably an overestimation, it’s better to state just “greater”. All of these changes are very small.

Response 10: “much greater” is corrected to “greater”.

Comment 11: Please confirm whether the crystal during bending remained within the plane or not. This may affect the measured angle from the recorded videos.

Response 11: We have analyzed bending motion in detail and confirmed that the crystal tip did not move out of focus during bending/straightening; this result indicates that the crystal bent within the plane and did not bend out of plane. The following sentence has been added in the “Natural vibration induced by the photothermal effect” section.

“Note that the crystal tip did not move out of focus during deformation, confirming that the crystal bent in the same plane and did not bend out of plane (Fig. 2a).”

Comment 12: p. 6. “infrared camera” and “thermography” have the same meaning. Please delete one of them.

Response 12: According to your suggestion, “thermography” is deleted in the page 6.

Comment 13: p. 7. How was the Young's modulus along the long direction measured?

Response 13: Young's modulus measurement is already described in Supplementary 4 (Supplementary Fig. 6). We omitted the description of Young's modulus measurement in the main text to avoid duplication.

Comment 14: Perhaps in the discussion the authors can add two comments for the readers who are not familiar with mechanical engineering:

(1) What would one need to do to increase the bending angle?

(2) What would one need to do to increase the frequency of vibration?

Response 14: The two comments mentioned above are already described in Fig. 4f and at the end of the first paragraph in "Actuation performance" results.

Comment 15: The quality of the images in panels d and c in Figure 1 should be increased.

Response 15: Fig 1c and d have been changed to the higher resolution images.

Comment 16: Please comment on how does the model for the photothermal bending relate to the recent quantitative model developed for photoinduced bending (Chem. Sci., 2018, 9; 2319–2335, Inorg. Chem. (2022) 61, 3573–3585) and include these related references.

Response 16: The previous quantitative models focus on the photoisomerisation-induced bending and are based on the chemical reaction. In contrast, the model in this study is based on the fundamental physical phenomena (the photothermal effect and the natural vibration) with the perspectives of material engineering. Therefore, the model for the photothermal bending is quite different and not related to the recent reported models. To clarify this, we added the following paragraph in front of the conclusion paragraph (the final paragraph of "Discussion" section) and the following references.

"We have successfully simulated photothermal-induced natural vibration based on FEA as shown in Simulation I and II. In the past decade, a few models have been proposed to reproduce photomechanical bending of crystals through photoisomerisation—chemical reactions of molecules—based on rigorous mathematical equations.^{29,30} The aim of these mathematical models is to understand the photoreaction of molecules in the crystalline state from the perspective of solid-state photochemistry. On the other hand, one of the objectives of this study is to reproduce high-speed bending of crystals through the photothermal effect and natural vibration—fundamental physical phenomena—and elucidate the mechanism of high-speed bending based on FEA models. Such FEA

models are expected to enable a deeper understanding of mechanical motions of crystals from the viewpoint of not only materials chemistry but also materials engineering.”

Ref. 29. Chizhik, S., Sidelnikov, A., Zakharov, B., Naumov, P. & Boldyreva, E. Quantification of photoinduced bending of dynamic molecular crystals: from macroscopic strain to kinetic constants and activation energies. *Chem. Sci.* 9, 2319–2335 (2018).

Ref. 30. Ahmed, E., Chizhik, S., Sidelnikov, A., Boldyreva, E. & Naumov, P. Relating Excited States to the Dynamics of Macroscopic Strain in Photoresponsive Crystals. *Inorg. Chem.* 61, 3573–3585 (2022).

Response to the comments from Reviewer 2

Comments: In this manuscript, Koshima et al. reported vibration of single crystals of 2,4-dinitroanisole induced by photoirradiation. The authors have been focusing on the crystalline compounds, which show large lattice parameter changes upon temperature change, and demonstrated the advantage of this strategy for development of photo-triggered actuating materials. In fact, the authors have already reported photothermally induced bending motion of single crystals, which also involved photoisomerization of component molecules and/or phase transitions during the mechanical motion. Unlike the previous systems developed by the authors, the crystalline system presented in this manuscript is the simplest one, where the component molecules only absorb UV-lights to generate the excited state, and neither photoisomerization nor phase transition occurs during the bending motion of the crystal. In addition, by taking advantage of pulsed light to cause the resonance with the natural frequency, amplification of the bending angle and fast actuating motion could be achieved. The methodology presented here seems to be applicable to wide variety of compounds, which allow for the photothermal process, and would give strong impacts on the field of organic functional materials. On the other hand, there are mainly three crucial issues which were not mentioned in the contents of the presented study.

Response: Thank you for your good evaluation and your important comments. We have carefully considered your comments and have revised the manuscript.

Comment 1: Organic crystals are three-dimensionally anisotropic due to the unsymmetric shape of the molecules. This means that physical parameters, such as Young's modulus, thermal diffusivity, as well as natural frequency, will have different values along different directions of the crystal. In the present study, the authors irradiated the (010) plane of a single crystal and bending of the crystal is described to occur parallel to the direction of the irradiated light, that is, perpendicular to the irradiated face. However, the shape of the crystal is indeed hexagonal, and (01-1) plane would also be irradiated.

If the Young's modulus in the direction perpendicular to the (010) plane is not the smallest, and/or if the thermal diffusivity is larger along the direction perpendicular to the (01-1) plane than that perpendicular to the (010) plane, the actual motion of the crystal may be more complexed than a simple linear bending motion. In fact, how large are the differences in the above-mentioned physical parameters among those along the different directions in a single crystal, and is it correct that the observed bending motion is simple as described in Fig. 5Aa?

Response 1: As your comment, physical parameters of the crystal have anisotropy. However, in this study, we measured the natural vibration when irradiated perpendicularly to only the (010) top surface of the hexagonal prismatic crystal. We did not measure the natural vibration when the (01-1) slanting surface was irradiated perpendicularly, due to the difficulty of measurement of thermal diffusivity. The relationship between crystal anisotropy and bending behavior will be the subject of future work.

At the present time the following has been clear. In this study, light was directed perpendicular to the (010) top surface so that the entire crystal was illuminated, so the (01-1) and (011) slanting surfaces were also illuminated from an oblique direction (Supplementary Fig. 13a). Therefore, heat is photothermally generated on the three surfaces. In this condition, the heat should be conducted only in the thickness direction. Thus, the bending should be in-plane bending (bend up/down) as shown in Figure 5a, and the photothermal bending and natural vibration depend on the Young's modulus in the length direction and thermal diffusivity in the thickness direction from Equations (1) and (4). In fact, the crystal tip remained in focus during observation of the crystal bending. This experimental fact is described by the additional sentence "Note that the crystal tip did not move out of focus during deformation, confirming that the crystal bent in the same plane and did not bend out of plane (Fig. 2a)." in the end of the first paragraph in "Natural vibration induced by the photothermal effect" section (yellow highlighted sentence, in accordance with the reviewer 1's comment).

Moreover, the experimental in-plane bending was confirmed by the FEA simulation. This is important to evaluate the simulated results. Thus, the following sentences are added to the second paragraph in "Simulation" subsection of "Discussion" section.

"Due to the one-directional temperature gradient in the thickness direction (Fig. 5b), the bending deformation was also confirmed to be in the vertical direction."

Comment 2: The authors claim that the bending speed and the resonated natural vibration in this system is the fastest among the previously reported systems. It is necessary to explain why such high performance could be achieved in the present system. In other words, does 2,4-dinitroanisole have any specific feature in the crystal packing which allows for such high performances? What is the most important parameter

for achieving these?

Response 2: The reason why the bending speed and resonated natural vibration of this system is the fastest among the previously reported systems is that the high Young's modulus of 2,4-dinitroanisole crystals creates the higher natural vibration frequency than frequencies of the previously reported bending.

The photothermal effect is a phenomenon by which thermal energy (heat) is generated very quickly at the picosecond level due to nonradiative deactivation of photoexcited state in primary photophysical process, creating the fast bending in millisecond order. In contrast, the photoisomerisation is slow at the nanosecond level chemical reaction that occur in secondary photochemical process, resulting in the slow bending in second to minute order. Natural vibration is triggered by the thermal load due to the photothermal heating in the crystal, thus it is as simultaneously occurred as the photothermally driven bending. However, it occurs repeatedly before the photothermal bending reaches its maximum as shown in Fig. 2d; that is the bending by natural vibration is caused faster than the bending by the photothermal effect itself. Indeed, natural vibration frequency is calculated based on Young's modulus, density, and size of crystal according to equation (1). Given that organic crystals have Young's modulus of 1–10 GPa, density of 1–2 g cm⁻³, and size of micro- to milli-meters, natural vibration frequency of organic crystals is calculated to be 100–10000 Hz, faster than photothermal-driven bending (up to 500 Hz at the present time). However, 2,4-dinitroanisole molecules have only weak π - π interaction in crystals, inducing the relatively smaller Young's modulus (1.65 GPa) and also not very high natural frequency among molecular crystals. Therefore, other molecular crystals would of course be expected to bend as fast as or faster than the 2,4-dinitroanisole crystal by natural vibration, which we intend to study it in the future.

The second paragraph in Actuation Performance was modified as follows:

[Before] “Next, we evaluated the energy conversion efficiency and the tip deflection speed of 1β crystals I–V and compared the results with other photomechanical crystals (Fig. 4B and Supplementary Table 3). The tip deflection speed v_t of the resonated natural vibration was in the range of 0.2–0.6 m s⁻¹ (black solid circle), which was more than ten times faster than the 0.01–0.03 m s⁻¹ (black open circle) of the non-resonated natural vibration. In addition, this speed is ten times faster than the 0.001–0.05 m s⁻¹ of the photothermally driven bending of the 1β crystal (red circle) and other crystals (red triangle,²¹ square,²² and rhombus²³), and two-to-three orders of magnitude faster than the 10⁻⁴–10⁻³ m s⁻¹ for photoisomerisation (blue triangle,²¹ square,²² and rhombus²³). To the best of our knowledge, the bending speed of the resonated natural vibration reported herein is the fastest of reported mechanically responsive crystals.”

[After] “Next, we evaluated the tip deflection speed of 1β crystals I–V and compared the results with other photomechanical crystals (Fig. 4B and Supplementary Table 3). The

tip deflection speed v_t of the resonated natural vibration was in the range of 0.2–0.6 m s⁻¹ (black solid circle), which was more than ten times faster than the 0.01–0.03 m s⁻¹ (black open circle) of the non-resonated natural vibration. The tip deflection speed (0.001 – 0.05 m s⁻¹) by the photothermal effect itself of the 1 β crystal (red circle) and other crystals (red triangle,²¹ square,²² and rhombus²³) is slightly slower to comparable to that of the non-resonated natural vibration, but still fast because the photothermal effect takes place at the picosecond level^{21,22} in the primary photophysical process. On the other hand, the tip deflection speed by photoisomerisation (blue triangle,²¹ square,²² and rhombus²³) is slow (0.0001 – 0.001 m s⁻¹) because photoisomerisation takes place at the nanosecond level^{21,22} in the secondary photochemical process. To the best of our knowledge, the bending speed of the resonated natural vibration reported herein is the fastest of reported mechanically responsive crystals. The high Young's modulus of 1 β crystals realised the highest natural frequency (700 Hz), contributing the fastest tip deflection speed."

Comment 3: What are the advantages of using the photothermal effects over IR irradiation? IR might be more energetically efficient than photoexcitation and would be applicable to much wide variety of compounds.

Response 3: Thank you for the good comment. We know of course that many organic crystals including this crystal have absorption in the infrared region. Unfortunately, however, we do not currently own an infrared light source. We intend to purchase an IR light source and examine the bending behaviour by IR irradiation (heating). When we obtain some results, we will report them elsewhere. We hope you understand that the value of this paper is the first finding of the photothermally induced natural vibration of crystals and its amplification by resonance.

Response to the comments from Reviewer 3

Comments: Publication is recommended after a revision. The first paragraph of the introduction "Natural vibration ... daily life" is general knowledge and can be deleted. The manuscript should be edited. Please define "by fast cooling". Please explain why the X-ray crystallographic analysis were carried at room temperature and not under LN2 conditions as is common. What is the origin of the strong absorption in the UV region (370 nm and a shoulder at 450 nm)? For how long are these crystals stable under UV light? PXRD measurements should be included. There is no need to justify this work with possible applications such as soft robotics. Data is shown for bend angles with UV on and UV off; what happens when the crystals are again exposed to UV? The authors should provide an indication of the errors of their experiments (temperature, bend angle and so on). What is a polarised photo (Figure 1b)? Although videos are provided, I would like to have a few clear photos included in the main text supporting cartoons as shown

in Fig 2c. A possible drawback of this work is the need for the formation of highly uniform crystals. As the authors showed, a different crystal shape results in different frequencies and bend angles. This effect limits the potential use of such crystals in possible applications. There is no need to justify this work with possible applications such as soft robotics. Instead, I would recommend that the authors address/discuss the issue of the need for development of crystal methods with control over crystal uniformity, dimensions and shape. I also recommend a discussion why these crystals have better properties than other systems.

Response: Thank you for your good evaluation and recommendation after a revision. According to your comments, we have revised the manuscript and the figures.

Comment 1: The first paragraph of the introduction “Natural vibration ... daily life” is general knowledge and can be deleted. The manuscript should be edited.

Response 1: The first paragraph of the introduction is general knowledge as you suggested. However, it is important for explaining how to create larger bending in terms of this research. The following sentence is added at the end of the first paragraph of the introduction to clarify the purpose of this paragraph.

“However, the natural vibration has not been paid attention in terms of material actuation.”

Comment 2: Please define “by fast cooling”.

Response 2: “Fast cooling” is defined as cooling for a short time using a cooling source. To avoid using the ambiguous word, the first sentence in “Physical properties of the crystals” section is corrected as follows:

[Before] “Colourless hexagonal rod-shaped 1β crystals were obtained by fast cooling of saturated methanol solution in a refrigerator at $\sim 5^{\circ}\text{C}$ (Fig. 1b).”

[After] “Colourless hexagonal rod-shaped **1** crystals were obtained from saturated methanol solution by cooling in a refrigerator at $\sim 5^{\circ}\text{C}$ for ~ 10 minutes (Fig. 1b).”

Comment 3: Please explain why the X-ray crystallographic analysis were carried at room temperature and not under LN2 conditions as is common.

Response 3: The reason the X-ray crystallographic analysis was conducted at room temperature is to obtain thermal expansion properties around room temperature; these thermal expansion properties are mandatory for bending simulation. On the other hand, information on crystal structures at LN2 conditions are not necessary in this study because the bending was observed at room temperature, not LN2 conditions.

Comment 4: What is the origin of the strong absorption in the UV region (370 nm and a shoulder at 450 nm)?

Response 4: The 2,4-dinitroanisole molecules in the crystals have the red-shifted absorption at around 370 nm due to stabilizing the excited energy by the π - π stacking (3.588 Å) of phenyl rings as shown in Fig.1d.

Comment 5: For how long are these crystals stable under UV light?

Response 5: We examined pulsed UV frequency dependence of bending for the same crystal over 18 hours (Fig. 3e) but no deformation or deterioration was observed, indicating good durability for at least 18 hours.

Comment 6: PXRD measurements should be included.

Response 6: PXRD is not performed in this study because the single X-ray crystallographic analysis gives all information on crystal structures, including lattice constants, face indices, and thermal expansion properties.

Comment 7: Data is shown for bend angles with UV on and UV off; what happens when the crystals are again exposed to UV?

Response 7: We investigated pulsed UV frequency dependence of high-speed bending by using up to 140 different frequencies, but the crystal did not get any fragile during the investigation.

Comment 8: The authors should provide an indication of the errors of their experiments (temperature, bend angle and so on).

Response 8: We observed crystal bending and temperature distribution change only once in each experiment. Thus, we cannot add indicators of the errors of experiments.

Comment 9: What is a polarized photo (Figure 1b)?

Response 9: A polarized photo means a photo taken by the polarized microscope under crossed Nicols. By using crossed Nicols state, we can distinguish each crystal domain because each domain has different polarization to display different brightness (colour) if polycrystalline state. In this research, this polarized photo revealed that the crystal in Fig. 1b was the single crystal because it had only one bright (white) domain. To clarify the description more, the figure legend in Fig. 1b is corrected as follows:

“b A photo of a 1 β single crystal under crossed Nicols”

Comment 10: Although videos are provided, I would like to have a few clear photos included in the main text supporting cartoons as shown in Fig 2c.

Response 10: The sequential snapshots before, upon, and after UV irradiation were added in Fig. 2a. The sentence explaining cartoons as shown in Fig 2c was also added in the main text as follows:

“Note that the crystal tip did not move out of focus during deformation, confirming that the crystal bent in the same plane and did not bend out of plane (Fig. 2a). In this study, crystal bending was evaluated using the bending angle, defined as the difference in tip displacement between the two most offset positions (Fig. 2c).”

Comment 11: A possible drawback of this work is the need for the formation of highly uniform crystals. As the authors showed, a different crystal shape results in different frequencies and bend angles. This effect limits the potential use of such crystals in possible applications. There is no need to justify this work with possible applications such as soft robotics. Instead, I would recommend that the authors address/discuss the issue of the need for development of crystal methods with control over crystal uniformity, dimensions, and shape.

Response 11: Thank you for the good comment. As you mentioned, what is the most significant point in this study is that we discovered high-speed bending of crystals by natural vibration for the first time, not on application. However, our group has been developed many light-driven crystals over the past decade, and our final goal is to apply these crystals to actuators and soft robots, and this research is no exception. Therefore, we mentioned the possible applications of crystals to actuators and soft robots in the main text.

As you commented, control of crystal size and shape is a very important research subject in crystal engineering. We have added the following sentence in the Actuation Performance section.

“However, precise control over uniformity, dimensions, and shape of crystals is still challenging; it is an essential requirement toward the development of crystal actuators with the desired output.”

Comment 12: I also recommend a discussion why these crystals have better properties than other systems.

Response 12: The reason why the bending speed and resonated natural vibration of this system is the fastest among the previously reported systems is that the high Young's modulus of 2,4-dinitroanisole crystals creates the higher natural vibration frequency than frequencies of the previously reported bending.

The photothermal effect is a phenomenon by which thermal energy (heat) is generated very quickly at the picosecond level due to nonradiative deactivation of photoexcited state in primary photophysical process, creating the fast bending in millisecond order. In contrast, the photoisomerisation is slow at the nanosecond level chemical reaction that occur in secondary photochemical process, resulting in the slow bending in second to minute order. Natural vibration is triggered by the thermal load due to the photothermal heating in the crystal, thus it is as simultaneously occurred as the photothermally driven bending. However, it occurs repeatedly before the photothermal bending reaches its maximum as shown in Fig. 2d; that is the bending by natural vibration is caused faster than the bending by the photothermal effect itself. Indeed, natural vibration frequency is calculated based on Young's modulus, density, and size of crystal according to equation (1). Given that organic crystals have Young's modulus of 1–10 GPa, density of 1–2 g cm⁻³, and size of micro- to milli-meters, natural vibration frequency of organic crystals is calculated to be 100–10000 Hz, faster than photothermal-driven bending (up to 500 Hz at the present time). However, 2,4-dinitroanisole molecules have only weak π - π interaction in crystals, inducing the relatively smaller Young's modulus (1.65 GPa) and also not very high natural frequency among molecular crystals. Therefore, other molecular crystals would of course be expected to bend as fast as or faster than the 2,4-dinitroanisole crystal by natural vibration, which we intend to study it in the future.

The second paragraph in Actuation Performance was modified as follows:

[Before] “Next, we evaluated the energy conversion efficiency and the tip deflection speed of **1 β** crystals I–V and compared the results with other photomechanical crystals (Fig. 4B and Supplementary Table 3). The tip deflection speed v_t of the resonated natural vibration was in the range of 0.2–0.6 m s⁻¹ (black solid circle), which was more than ten times faster than the 0.01–0.03 m s⁻¹ (black open circle) of the non-resonated natural vibration. In addition, this speed is ten times faster than the 0.001–0.05 m s⁻¹ of the photothermally driven bending of the **1 β** crystal (red circle) and other crystals (red triangle,²¹ square,²² and rhombus²³), and two-to-three orders of magnitude faster than the 10⁻⁴–10⁻³ m s⁻¹ for photoisomerisation (blue triangle,²¹ square,²² and rhombus²³). To the best of our knowledge, the bending speed of the resonated natural vibration reported herein is the fastest of reported mechanically responsive crystals.”

[After] “Next, we evaluated the tip deflection speed of **1 β** crystals I–V and compared the results with other photomechanical crystals (Fig. 4B and Supplementary Table 3). The tip deflection speed v_t of the resonated natural vibration was in the range of 0.2–0.6 m s⁻¹ (black solid circle), which was more than ten times faster than the 0.01–0.03 m s⁻¹ (black open circle) of the non-resonated natural vibration. The tip deflection speed (0.001 – 0.05 m s⁻¹) by the photothermal effect itself of the **1 β** crystal (red circle) and other crystals (red triangle,²¹ square,²² and rhombus²³) is slightly slower to comparable to that of the non-resonated natural vibration, but still fast because the photothermal effect takes place at the picosecond level^{21,22} in the primary photophysical process. On the other

hand, the tip deflection speed by photoisomerisation (blue triangle,²¹ square,²² and rhombus²³) is slow ($0.0001 - 0.001 \text{ m s}^{-1}$) because photoisomerisation takes place at the nanosecond level^{21,22} in the secondary photochemical process. To the best of our knowledge, the bending speed of the resonated natural vibration reported herein is the fastest of reported mechanically responsive crystals. The high Young's modulus of **1 β** crystals realised the highest natural frequency (700 Hz), contributing the fastest tip deflection speed."

Reviewers' Comments:

Reviewer #1:

Remarks to the Author:

The authors have diligently addressed all comments, and as a result, their manuscript has been improved. I recommend publication in its present form.

Reviewer #2:

Remarks to the Author:

The responses from the authors were very sincere which addressed most of the concerns raised by this reviewer. Also, the manuscript has been revised to include appropriate additional explanations and discussions to support the author's claims. Therefore, this manuscript is basically recommended for publication in Nature Communications.

However, the author's response leaves some uncertainty about the reason why 2,4-dinitroanisole showed such high performance. At the introductory part in page 4, the authors seem to have focused on the relatively large thermal expansion coefficient of 1β . Meanwhile, equation (1) in page 7 suggests an importance of Young's modulus for achieving high frequency. Indeed, in the Response 2 to this reviewer, the authors explained that high Young's modulus of 2,4-dinitroanisole is the reason for its high natural vibration frequency. However, at the same time, the authors also wrote that the Young's modulus of 2,4-dinitroanisole is rather small compared with usual organic crystals. These explanations are confusing. Why did the authors explain that the Young's modulus for 2,4-dinitroanisole (1β) is high? What is the basic for the comparison?

Reviewer #3:

Remarks to the Author:

Although the authors made some changes to the manuscript according to my comments, I do think that PXRD measurements should be included. It is a standard method and can provide information of the bulk.

Also bothersome is the lack of error analysis: The authors should provide an indication of the errors of their experiments (temperature, bend angle and so on). According to their reply, they cannot provide such data "We observed crystal bending and temperature distribution change only once in each experiment. Thus, we cannot add indicators of the errors of experiments."

In my opinion, these two issues have to be resolved before publication can be recommended.

Point-by-point response to the reviewers' comments

We thank the reviewers for the good evaluation and the additional comments of our paper and recommendation to publish after revisions. We have thoroughly revised the manuscript according to the reviewers' comments.

Response to the comments from Reviewer 1

Comment:

The authors have diligently addressed all comments, and as a result, their manuscript has been improved. I recommend publication in its present form.

Response:

Thank you for your good evaluation.

Response to the comments from Reviewer 2

Comment:

The responses from the authors were very sincere which addressed most of the concerns raised by this reviewer. Also, the manuscript has been revised to include appropriate additional explanations and discussions to support the author's claims. Therefore, this manuscript is basically recommended for publication in Nature Communications.

Response:

Thank you for your good evaluation.

Comment 1:

However, the author's response leaves some uncertainty about the reason why 2,4-dinitroanisole showed such high performance. At the introductory part in page 4, the authors seem to have focused on the relatively large thermal expansion coefficient of 1β . Meanwhile, equation (1) in page 7 suggests an importance of Young's modulus for achieving high frequency. Indeed, in the Response 2 to this reviewer, the authors explained that high Young's modulus of 2,4-dinitroanisole is the reason for its high natural vibration frequency. However, at the same time, the authors also wrote that the Young's

modulus of 2,4-dinitroanisole is rather small compared with usual organic crystals. These explanations are confusing. Why did the authors explain that the Young's modulus for 2,4-dinitroanisole (1β) is high? What is the basic for the comparison?

Response 1:

Thank you for pointing out the uncertainty. Initially, our research objective was to create large bending due to the photothermal effect, so we focused on the 1β crystal with a large thermal expansion coefficient, as described at the Introductory part in page 4. However, while observing the crystal bending due to the photothermal effect upon UV light irradiation, we have unexpectedly discovered for the first time that the small, fast natural vibration occurs while accompanied by the large photothermal bending. To clarify this process, we have revised the last sentence and the first sentence of the third and fourth paragraphs, respectively, at the Introductory part in page 4 as follows.

[Before] ---- One strategy to enhance photothermal bending is to focus on crystals with a large thermal expansion coefficient.

Herein, we report that β -phase crystal (1β)²⁴ of 2,4-dinitroanisole (**1**) (Fig. 1a), which has relatively large thermal expansion coefficient, exhibit small but fast natural vibration associated with larger photothermally driven bending during ultraviolet (UV) laser irradiation. ----

[After] ---- One strategy to enhance photothermal bending is to focus on crystals with a large thermal expansion coefficient. Hence β -phase crystal (1β)²⁴ of 2,4-dinitroanisole (**1**) (Fig. 1a), which has relatively large thermal expansion coefficient, was used in this study.

While observing the bending of 1β crystals due to the photothermal effect upon ultraviolet (UV) light irradiation, we have unexpectedly discovered for the first time that small, fast natural vibration occurs while accompanied by a large photothermal bending.

According to equation (1) in page 7, Young's modulus is an important parameter to achieve fast natural vibration. The Young's modulus of the 1β crystal is 1.65 GPa. On the other hand, Young's moduli of usual organic crystals are in the range of 1 to 25 GPa [Ref. 17], which means that the Young's modulus of the 1β crystal is small among usual organic crystals. In contrast, the Young's moduli of soft materials such as polymers and gels are small, on the order of MPa and kPa [Ref. 17], which means that the Young's modulus of the 1β crystal is large when compared to polymers and gels. We revised the last sentence of the second paragraph in Actuation performance as follows.

[Before] The high Young's modulus of 1β crystals realised the highest natural frequency (700 Hz), contributing the fastest tip deflection speed.

[After] The larger Young's modulus of the 1β crystal than those of polymers and gels realised the high natural frequency (700 Hz), contributing the fast tip deflection speed.

Response to the comments from Reviewer 3

Comment 1:

Although the authors made some changes to the manuscript according to my comments, I do think that PXRD measurements should be included. It is a standard method and can provide information of the bulk.

Response 1:

The PXRD pattern obtained from the X-ray crystallographic data was added in Supplementary Fig. 1b.

Comment 2:

Also bothersome is the lack of error analysis: The authors should provide an indication of the errors of their experiments (temperature, bend angle and so on). According to their reply, they cannot provide such data "We observed crystal bending and temperature distribution change only once in each experiment. Thus, we cannot add indicators of the errors of experiments."

Response 2:

We surely understand the reviewer's opinion that we should provide an error indication. In fact, we already evaluated the errors of unit cell parameters changes in Fig. 1e and Supplementary Table 2 as well as the error of Young's modulus in Supplementary Table 3. However, as explained in the previous revision, we did not measure crystal bending and temperature distribution change in several times, they were measured only one time. Thus, we could not evaluate errors of experiments. Instead of this, we could conduct an error analysis for the steady oscillating region (133 cycles during 850 –1190 ms) in 390 Hz resonated natural vibration of crystal III (Fig. 3a); the average bending angle is 2.3° with the standard error of 0.003° , showing quite small error. We revised the fourth sentence of the first paragraph in Resonance amplification of a natural vibration, and

added the new sentence about the error as follows.

[Before] Then, the resonated bending angle gradually decreased as the surface temperature increased and reached the steady bending angle of 2.3° when the surface temperature reached the constant 40.8°C after 850 ms under pulsed UV light.

[After] Then, the resonated bending angle gradually decreased as the surface temperature increased. *When the surface temperature reached a constant 40.8°C after 850 ms under pulsed UV light, the resonated bending angle reached a steady value of 2.3° and maintained the angle with a quite small standard error of 0.003° during 850–1190 ms (133 cycles).*

Comment:

In my opinion, these two issues have to be resolved before publication can be recommended.

Response:

We have resolved the two issues in the above responses.